# Stochastic Solutions for Linear Inverse Problems using the Prior Implicit in a Denoiser

**Zahra Kadkhodaie**
Center for Data Science,
New York University
zk388@nyu.edu

**Eero P. Simoncelli**
Center for Neural Science, and
Courant Inst. of Mathematical Sciences,
New York University
Flatiron Institute, Simons Foundation
eero.simoncelli@nyu.edu

## Abstract

Deep neural networks have provided state-of-the-art solutions for problems such as image denoising, which implicitly rely on a prior probability model of natural images. Two recent lines of work – Denoising Score Matching and Plug-and-Play – propose methodologies for drawing samples from this implicit prior and using it to solve inverse problems, respectively. Here, we develop a parsimonious and robust generalization of these ideas. We rely on a classic statistical result that shows the least-squares solution for removing additive Gaussian noise can be written directly in terms of the gradient of the log of the noisy signal density. We use this to derive a stochastic coarse-to-fine gradient ascent procedure for drawing high-probability samples from the implicit prior embedded within a CNN trained to perform blind denoising. A generalization of this algorithm to constrained sampling provides a method for using the implicit prior to solve any deterministic linear inverse problem, with no additional training, thus extending the power of supervised learning for denoising to a much broader set of problems. The algorithm relies on minimal assumptions and exhibits robust convergence over a wide range of parameter choices. To demonstrate the generality of our method, we use it to obtain state-of-the-art levels of unsupervised performance for deblurring, super-resolution, and compressive sensing.

## 1 Introduction

Many problems in image processing and computer vision rely, explicitly or implicitly, on prior probability models. Describing the full density of natural images is a daunting problem, given the high dimensionality of the signal space. Traditionally, models have been developed by combining assumed symmetry properties (e.g., translation-invariance, dilation-invariance), with simple parametric forms (e.g., Gaussian, exponential, Gaussian mixtures), often within pre-specified transformed coordinate systems (e.g., Fourier transform, multi-scale wavelets). While these models have led to steady advances in problems such as denoising (e.g., [1–7]), they are too simplistic to generate complex features that occur in our visual world, or to solve more demanding statistical inference problems.

In recent years, nearly all problems in image processing and computer vision have been revolutionalized by the use of deep Convolutional Neural Networks (CNNs). These networks are generally optimized in supervised fashion to obtain a direct input-output mapping for a specific task. This approach does not explicitly rely on a known prior, and offers performance far superior to prior-based methods. The downside, however, is that the learned mappings are intertwined with the task for which they are optimized, and in most cases require training a separate network for each new application. In contrast, a prior probability model can provide a universal substrate for solving inference

35th Conference on Neural Information Processing Systems (NeurIPS 2021).

problems. The superior performance of CNNs suggests that they embed, implicitly, sophisticated prior knowledge of images. These implicit priors arise from a combination of the distribution of the training data, the architecture of the network [8], regularization terms included in the optimization objective, and the optimization algorithm.

Here, our goal is to extract the implicit prior from a network trained for denoising and use it to solve other inverse problems without further training. We choose denoising not because it is of particular importance or interest, but because we can make the relationship between mapping of a denoiser and a prior explicit. We combine the advantages of prior-based and mapping-based approaches, deriving a general algorithm for solving linear inverse problems using the prior implicit in a trained denoiser. We start with a result from classical statistics [9] that states that a denoiser that aims to minimize squared error of images corrupted by additive Gaussian noise may be interpreted as computing the gradient of the log of the density of noisy images. This result is related to Score Matching [10], but provides a more direct relationship between least-squares optimal denoising and the embedded prior [11, 12]. We develop a stochastic ascent algorithm that uses this denoiser-estimated gradient to draw high-probability samples from the embedded prior. Importantly, we use a *blind* denoiser that can handle noise contamination of unknown amplitude, which provides a means of adaptively controlling the gradient step sizes and the amplitude of injected noise, enabling robust and efficient convergence. We then modify the algorithm to incorporate constraints arising from any deterministic linear measurement of an image. The resulting procedure generates high-probability samples from the prior conditioned on the measurements, thus providing a general stochastic solution for any deterministic linear inverse problem. We demonstrate that our method produces visually high-quality results in recovering missing pixels, and state-of-the-art levels of unsupervised performance on superresolution, deblurring and compressive sensing.[1] Earlier versions of this work were presented in [13].

This work is closely related to two lines of research. Nearly a decade ago, a strategy known as Plug-and-Play (P&P) was proposed for using a denoiser as a regularizer in solving other inverse problems [14], and a number of recent extensions have used this concept to develop MAP solutions for linear inverse problems [15–22, 12]. Generally, the objective is decoupled into data fidelity and regularization terms, introducing a slack variable for use in a proximal optimization algorithm (e.g., ADMM). The proximal operator of the regularization term is interpreted as the MAP solution of a denoising problem, and is replaced by a denoiser. Of particular relevance to our work, recent publications have proven convergence of such algorithms when used in conjunction with MMSE denoisers [23, 24]. A parallel line of research has focused on the use of generative models based on Score Matching [25–30]. The connection between Score Matching [10] and denoising autoencoders [31] was first shown in [32], by proving that the training criterion of a denoising autoencoder is equivalent to matching the score of the model and a Parzan density estimate of the data. Most recently, this idea has been used as the basis for an MCMC algorithm for sampling from the prior implicit in a CNN denoiser [29]. In Section 4 we elaborate on how these methods are related to our results.

## 1.1 Image priors, manifolds, and noisy observations

Digital photographic images lie in a high-dimensional space ($\mathbb{R}^N$, where $N$ is the number of pixels), and simple thought experiments suggest that they are concentrated on or near low-dimensional manifolds whose local coordinates represent continuous deformations and intensity variations. In contrast, images generated with random pixels are almost always feature and content free, and thus not considered to be part of this manifold. We can associate with this manifold a prior probability model, $p(x)$, by assuming that images within the manifold have constant or slowly-varying probability, while unnatural or distorted images (which lie off the manifold) have low or zero probability. Suppose we make a noisy observation of an image, $y = x + z$, where $x \in R^N$ is the original image drawn from $p(x)$, and $z \sim \mathcal{N}(0, \sigma^2 I_N)$ is a sample of Gaussian white noise. The observation density $p(y)$ is related to the prior $p(x)$ via marginalization:

$$p(y) = \int p(y|x)p(x)dx = \int g(y-x)p(x)dx, \tag{1}$$

where $g(z)$ is the Gaussian noise distribution. Equation (1) is in the form of a convolution, and thus $p(y)$ is a Gaussian-blurred version of the signal prior, $p(x)$. Moreover, the family of observation

---

[1] A software implementation of the sampling and linear inverse algorithms is available at `https://github.com/LabForComputationalVision/universal_inverse_problem`

densities over different noise variances, $p_\sigma(y)$, forms a Gaussian scale-space representation of the prior [33, 34], analogous to the temporal evolution of a diffusion process.

## 1.2 Least squares denoising and CNNs

Given a noisy observation, $y$, the minimum mean squared error (MMSE) estimate of the true signal is well known to be the conditional mean of the posterior density:

$$\hat{x}(y) = \int x p(x|y) dx = \int x \frac{p(y|x)p(x)}{p(y)} dx \tag{2}$$

The structure of the equation mirrors the traditional approach to the problem: one chooses a prior probability model, $p(x)$, combines it with a likelihood function describing the noisy measurement process, $p(y|x)$, and solves. Modern denoising solutions, on the other hand, are often based on supervised learning of a direct mapping from noisy to denoised images. One expresses the estimation function (as opposed to the prior) in parametric form, and sets the parameters by minimizing the denoising MSE over a large training set of example signals and their noise-corrupted counterparts [35–38]. Current state-of-the-art denoising results using CNNs obtained with this supervised approach are far superior to results of previous methods [39–41]. Recent analysis of these networks demonstrates that when they are trained to handle a broad range of noise levels, they perform an approximate projection onto a low-dimensional subspace [42]. In our context, we interpret this subspace as a tangent hyperplane of the image manifold.

## 1.3 Exposing the implicit prior through Empirical Bayes estimation

Trained CNN denoisers contain detailed prior knowledge of image structure, but Eq. (2) suggests that it is embedded within a high-dimensional integral. How can we make use of this implicit prior? Recent results have derived relationships between Score Matching density estimates and denoising, and have used these relationships to make use of implicit prior information [43, 44, 29, 45]. Here, we exploit a more direct but less-known result from the literature on Empirical Bayesian estimation. The idea was introduced in [46], extended to the case of Gaussian additive noise in [9] (see also [12]), and generalized to many other measurement models [11]. In the case of additive Gaussian noise, one can rewrite the estimator of Eq. (2) as:

$$\hat{x}(y) = y + \sigma^2 \nabla_y \log p(y). \tag{3}$$

The proof is relatively straightforward. The gradient of the observation density of Eq. (1) is:

$$\nabla_y\, p(y) = \frac{1}{\sigma^2} \int (x - y) g(y - x) p(x) dx = \frac{1}{\sigma^2} \int (x - y) p(y, x) dx.$$

Multiplying both sides by $\sigma^2/p(y)$ and separating the right side into two terms gives:

$$\sigma^2 \frac{\nabla_y\, p(y)}{p(y)} = \int x p(x|y) dx - \int y p(x|y) dx = \hat{x}(y) - y.$$

Rearranging terms and using the chain rule to compute the gradient of the log gives Eq. (3). This remarkable result re-expresses the integral over the prior and likelihood of Eq. (2) in terms of a *gradient*. Note that 1) the relevant density is not the prior, $p(x)$, but the noisy *observation density*, $p(y)$; 2) the gradient is computed on the *log* density (the associated "energy function"); and 3) the gradient adjustment is *not* iterative - the estimate is achieved in a single step, and holds for any noise level, $\sigma$.

## 2 Drawing high-probability samples from the implicit prior

Suppose we wish to draw a sample from the prior implicit in a denoiser. Equation (3) allows us to generate an image proportional to the gradient of $\log p(y)$ by computing the denoiser residual, $f(y) = \hat{x}(y) - y$. Song and Ermon [29] developed a Markov chain Monte Carlo (MCMC) scheme, combining gradient steps derived from Score Matching and injected noise in a Langevin sampling algorithm to draw samples from a sequence of densities $p_\sigma(y)$, while reducing $\sigma$ in a sequence of discrete steps, each associated with an appropriately trained denoiser. In contrast, starting from a random initialization, $y_0$, we aim to find a *high-probability* image (i.e., an image from the manifold) using a more direct and efficient stochastic gradient ascent procedure.

## 2.1 Unconstrained sampling algorithm

We compute gradients using the residual of a  universal blind CNN denoiser, which automatically estimates and adapts to each noise level. On each iteration, the algorithm takes a small step in the direction specified by the denoiser, moving toward the image manifold and thereby reducing the amplitude of the effective noise. Under the interpretation that the denoiser performs a projection onto the current approximation of the image manifold, this noise reduction occurs in the subspace orthogonal to that manifold, and noise components parallel to the manifold are retained. As the effective noise decreases, the observable dimensionality of the image manifold increases [42], enabling the synthesis of detailed image content. Since the family of observation densities, $p_\sigma(y)$ forms a scale-space representation of $p(x)$, the algorithm may be viewed as a form of coarse-to-fine optimization [47-50]. Assuming the step sizes are adequately controlled, the procedure will converge to a local optimum of the implicit prior - i.e., a point on the manifold. Figure 7 provides a visualization of this process in two dimensions.

Each iteration operates by taking a deterministic step in the direction of the gradient (as obtained from the denoising function) and injecting some additional noise:

$$y_t = y_{t-1} + h_t f(y_{t-1}) + \gamma_t z_t, \tag{4}$$

where $f(y) = \hat{x}(y) - y$ is the residual of the denoising function, which is proportional to the gradient of $\log p(y)$, from Eq. (3). The parameter $h_t \in [0, 1]$ controls the fraction of the denoising correction that is taken, and $\gamma_t$ controls the amplitude of a sample of white Gaussian noise, $z_t \sim \mathcal{N}(0, I)$. The purpose of injecting noise is two-fold. First, from an optimization perspective, it allows the method to avoid getting stuck in local maxima. Second, it allows stochastic exploration of the manifold, yielding a more diverse (higher entropy) family of solutions.  The effective noise variance of image $y_t$ is:

$$\sigma_t^2 = (1 - h_t)^2 \sigma_{t-1}^2 + \gamma_t^2, \tag{5}$$

where the first term is the variance of the noise remaining after the denoiser correction (assuming the denoiser is perfect), and the second term is the variance arising from the injected noise. The assumption of perfect denoising is an idealization, but we show empirically (Fig. 1) that the algorithm converges reliably, with error levels falling as predicted by Eq. (5) or faster, across different settings of $\beta$ and $h_0$.

To ensure convergence, we require the effective noise variance on each time step to be reduced, despite the injection of additional noise. For this purpose, we introduce a parameter $\beta \in [0, 1]$ to control the proportion of injected noise ($\beta = 1$ indicates no noise), and enforce the convergence by requiring that:

$$\sigma_t^2 = (1 - \beta h_t)^2 \sigma_{t-1}^2. \tag{6}$$

Combining this with Eq. (5) yields an expression for $\gamma_t$ in terms of $h_t$:

$$\gamma_t^2 = \left[(1 - \beta h_t)^2 - (1 - h_t)^2\right] \sigma_{t-1}^2 = \left[(1 - \beta h_t)^2 - (1 - h_t)^2\right] \|f(y_{t-1})\|^2 / N,$$

where the second equation assumes that the magnitude of the denoising residual provides a good estimate of the effective noise standard deviation, as was found in [42]. This allows the denoiser to adaptively control the gradient ascent step sizes, reducing them as the $y_t$ approaches the manifold (see Fig. 7). This automatic adjustment results in efficient and reliable convergence, as demonstrated empirically in Fig. 1. Our initial implementation with a small constant fractional step size $h_t = h_0$ produced high quality results, but required many iterations. Intuitively, step sizes that are a fixed proportion of the distance to the manifold lead to exponential decay - a form of Zeno's paradox. To accelerate convergence, we introduced a schedule for increasing the step size proportion, starting

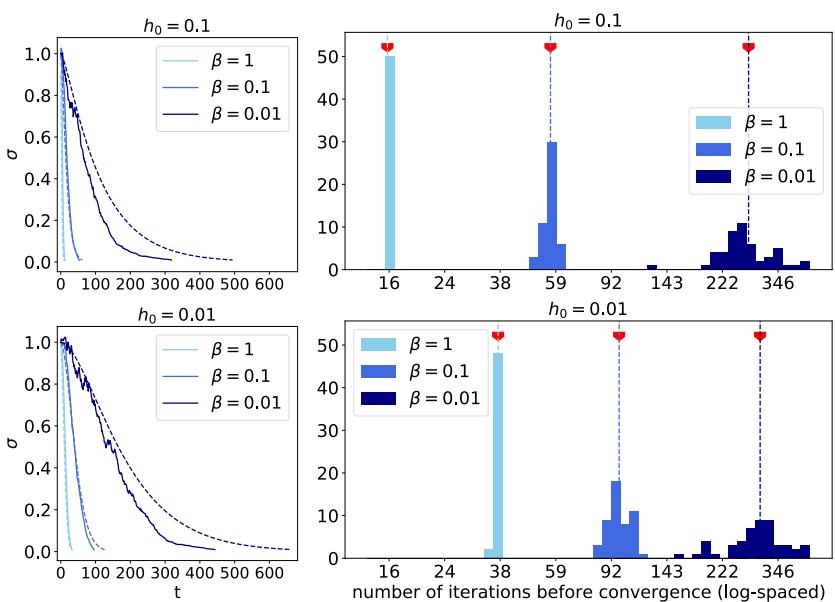

Figure 1: Convergence of sampling algorithm, quantified in terms of the effective noise standard deviation $\sigma = \frac{||d_t||}{\sqrt{N}}$ for three different values of $\beta$ and two valus of $h_0$. **Left.** Convergence of single examples (solid curves) is well-behaved and efficient in all cases. Dashed curves indicate the convergence predicted from the formulation of the algorithm: $\sigma_t = (1 - \beta h_t)\sigma_{t-1}$. For $\beta = 1$ (no injected noise), the empirical convergence closely approximates the prediction. For larger amounts of injected noise (smaller $\beta$), convergence is slower, but faster than predicted. **Right.** Distribution of number of iterations before convergence to a criterion level of $\sigma = 0.01$ for 50 images. Red symbols indicate average values for each $\beta$ and $h_0$.

from $h_0 \in [0, 1]$. The sampling process is summarized in Algorithm 1, and is provided as a block diagram in Fig. 6.

---

**Algorithm 1:** Coarse-to-fine stochastic ascent method for sampling from the implicit prior of a denoiser, using denoiser residual $f(y) = \hat{x}(y) - y$.

---

parameters: $\sigma_0, \sigma_L, h_0, \beta$
initialization: $t = 1$, draw $y_0 \sim \mathcal{N}(0.5, \sigma_0^2 I)$
**while** $\sigma_{t-1} \geq \sigma_L$ **do**
$\quad h_t = \frac{h_0 t}{1 + h_0(t-1)}$;
$\quad d_t = f(y_{t-1})$;
$\quad \sigma_t^2 = \frac{||d_t||^2}{N}$;
$\quad \gamma_t^2 = \left((1 - \beta h_t)^2 - (1 - h_t)^2\right)\sigma_t^2$;
$\quad$ Draw $z_t \sim \mathcal{N}(0, I)$;
$\quad y_t \leftarrow y_{t-1} + h_t d_t + \gamma_t z_t$;
$\quad t \leftarrow t + 1$
**end**

---

### 2.2 Image synthesis examples

For the denoiser, we used BF-CNN [42], a bias-free variant of DnCNN [39]. We obtained similar results (not shown) using other CNN architectures described in [42], including Recurrent-CNN, Dense-Net, and truncated U-Net. We trained this network on three different datasets: $40 \times 40$ patches cropped from Berkeley segmentation training set [51], in color and grayscale, and MNIST dataset [52] (see Appendix A for further details). We chose parameters $\sigma_0 = 1, \sigma_L = 0.01$, and $h_0 = 0.01$. Figure 2 provides visualization of two example trajectories, and diversity of samples. Additional visual examples, obtained with different levels of $\beta$, are shown in Appendix D.

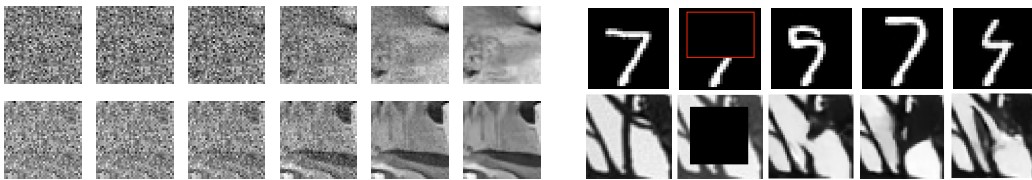

Figure 2: **Left.** Visualization of sampling algorithm trajectories. Each row shows a sequence of images, $y_t, t = 1, 9, 17, 25, \ldots$, from the iterative sampling procedure, with different initializations, $y_0$, and no added noise ($\beta = 1$), demonstrating the way that the algorithm amplifies and "hallucinates" structure found in the initial (noise) images. **Right.** Sampling diversity. Inpainting examples generated using two BF-CNN denoisers. First column: original images. Second column: partially measured images, with missing block . Right three columns: Restored examples.

## 3 Solving deterministic linear inverse problems using the implicit prior

Many applications in signal processing can be expressed as deterministic linear inverse problems - deblurring, super-resolution, estimating missing pixels (e.g., inpainting), and compressive sensing are all examples. Given a set of linear measurements of an image, $x^c = M^T x$, where $M$ is a low-rank measurement matrix, one attempts to recover the original image. In Section 2, we developed a stochastic gradient-ascent algorithm for obtaining a high-probability sample from $p(x)$. Here, we generalize this algorithm to solve for a high-probability sample from the conditional density $p(x|M^T x = x^c)$. Geometrically, constrained sampling corresponds to drawing points which sit at the intersection of the image manifold and the constrained hyperplane (see Fig. 11). Note that for these problems, the injection of noise ($\beta < 1$) is particularly important, because points on the intersection are not necessarily the closest points to the initial image.

### 3.1 Constrained sampling algorithm

Consider the distribution of a noisy image, $y$, conditioned on the linear measurements, $x^c = M^T x$. Without loss of generality, we assume the columns of the matrix M are orthogonal unit vectors.[2] We project $y$ onto two complementary subspaces spanned by the measurement matrix and its orthogonal complement $\bar{M}$. We write the conditional density of the noisy image conditioned on the linear measurement as

$$p(y|x^c) = p(y^c, y^u|x^c) = p(y^u|y^c, x^c)p(y^c|x^c) = p(y^u|x^c)p(y^c|x^c)$$

where $y^c = M^T y$, $y^u = \bar{M}^T y$. The last equality is obtained by considering that $y^c$ is equal to $x^c$ plus independent Gaussian noise. So given $x^c$, $y^c$ does not provide any additional information about $y^u$. That is, $y^u$ is independent of $y^c$ when conditioned on $x^c$. As with the algorithm of Section 2, we wish to obtain a local maximum of this function using stochastic coarse-to-fine gradient ascent. Applying the operator $\sigma^2 \nabla \log(\cdot)$ yields

$$\sigma^2 \nabla_y \log p(y|x^c) = \sigma^2 \nabla_y \log p(y^u|x^c) + \sigma^2 \nabla_y \log p(y^c|x^c)$$

The second term is the gradient of the log of the observation noise distribution that lies within the measurement subspace (column space of M). For Gaussian noise with variance $\sigma^2$, it reduces to $M(y^c - x^c)$. The first term is the gradient of log of the noisy image distribution in the subspace orthogonal to the measurement subspace, conditioned on the measurements. This can be computed by projecting the measurement subspace out of the full gradient given by the denoiser residual. Specifically, we project $f(y)$ onto the orthogonal complement of $M$ using the matrix $I - MM^T$. Combining these gives:

$$\sigma^2 \nabla_y \log p(y) = (I - MM^T)\sigma^2 \nabla_y \log p(y) + M(x^c - y^c)$$
$$= (I - MM^T)f(y) + M(x^c - M^T y). \quad (7)$$

Thus, we see that the gradient of the conditional density is partitioned into two orthogonal components, capturing the gradient of the (log) noisy density, and the deviation from the constraints, respectively.

---

[2]If not, we can re-parameterize to an equivalent constraint using the SVD. In this case, $M$ is the pseudo-inverse of $M^T$, and the matrix $MM^T$ projects an image onto the measurement subspace.

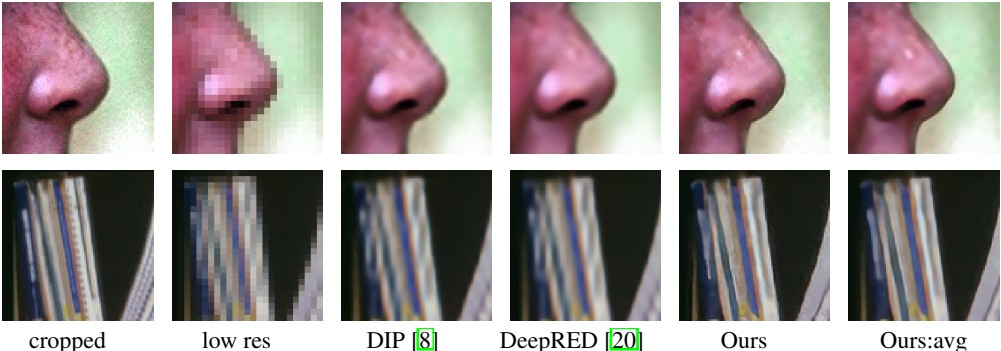

| cropped | low res | DIP [8] | DeepRED [20] | Ours | Ours:avg |

Figure 3: Spatial super-resolution. First column shows cropped portion of original images from Set14 (face: $272 \times 272$ and, Barbara: $720 \times 576$). The algorithm can be applied to images of any resolution - we show cropped portions to facilitate visual inspection. Second column shows cropped portion with resolution reduced by averaging over 4x4 blocks (dimensionality reduction to 6.25%). Next three columns show reconstruction results obtained using DIP [8], DeepRED [20], and our method. In all cases, our method produces an image that is sharper with less noticeable artifacts (e.g., note blocking/aliasing artifacts along diagonal contours of lower image). The last column shows an average over 10 samples obtained by our method.

To draw a high-probability sample from $p(x|x^c)$, we use the same algorithm described in Section 2, substituting Eq. (7) for the deterministic update vector, $d_t$ (see Algorithm 2 in Appendix). Note that without any measurements (i.e., $M = 0$) Algorithm 2 reduces to Algorithm 1.

## 3.2   Linear inverse examples

We evaluate our method on three linear inverse problems (and provide two additional problems in Appendix E). The same algorithm and parameters are used on all problems - only the measurement matrix $M$ and measured values $M^T x$ are altered. In particular, as in section 2.2, we used BF-CNN [42], and chose parameters $\sigma_0 = 1, \sigma_L = 0.01, h_0 = 0.01, \beta = 0.01$. For each example, we show original images ($x$), the direct least-squares reconstruction ($MM^T x$), and restored images. Subjective assessment of perceptual quality is particularly important in cases where the measurement matrix is of very low rank, and the distribution of solutions is diverse (the synthesis examples of the previous section correspond to the limiting case, with measurements of rank zero). We also provide numerical comparisons with other unsupervised methods, in terms of both PSNR and SSIM (an approximate measure of perceptual quality). Since our method is stochastic, we provide standard deviations of these performance values across 10 realizations.

**Spatial super-resolution.** Here, one aims to reconstruct a high resolution image from a low resolution (i.e. downsampled) image. Downsampling is typically performed after lowpass filtering, and the downsampling factor and filter kernel determine the measurement model, $M$. Here, we use a $4 \times 4$ constant filter, and $4 \times 4$ downsampling (i.e., measurements are averages over non-overlapping blocks). We compare to two recent unsupervised methods Deep image Prior (DIP) [8] and DeepRED [20]. DIP chooses a random input vector, and adjusts the weights of a CNN to minimize the mean square error between the output and the corrupted image. Regularization by denoising (RED) is a recent successful method closely related to P&P [15]. DeepRED [20] combines DIP and RED, obtaining better performance than either method alone.

Inspection of results on two example images demonstrates that our method produces results that are sharper with less noticeable artifacts (Fig. 3). Despite this, the PSNR and SSIM values are slightly worse than both DIP and DeepRED (Table 1). These can be improved by averaging over realizations (last column of Table 1), producing superior PSNR and SSIM values at the expense of some blurring (last column of Fig. 3). This is expected: the algorithm produces high-probability samples of the prior subject to the measurement constraint, but the least-squares optimal solution is the mean of the posterior distribution. If the samples are drawn from a curved manifold, their average (a convex combination of those points) will lie off the manifold. Finally, note that our method is more than two orders of magnitude faster than either DIP or DeepRED (bottom row, Table 1).

Table 1: Spatial super-resolution performance over Set5 (top 2 rows) and Set14 (second 2 rows). Values indicate YCbCr-PSNR (SSIM). Last row shows average Set14 runtime on a DGX GPU.

|  | $MM^Tx$ | DIP [8] | DeepRED [20] | Ours±std | Ours:avg |
|---|---|---|---|---|---|
| 4:1 | 26.35 (0.826) | 30.04 (0.902) | 30.22 (0.904) | 29.47±0.09 (0.894±0.001) | **31.20 (0.913)** |
| 8:1 | 23.02 (0.673) | 24.98 (0.760) | 24.95 (0.760) | 25.07±0.13 (0.767±0.003) | **25.64 (0.792)** |
| 4:1 | 24.65 (0.765) | 26.88 (0.815) | 27.01 (0.817) | 26.56±0.09 (0.808±0.001) | **27.14 (0.826)** |
| 8:1 | 22.06 (0.628) | 23.33 (0.685) | 23.34 (0.685) | 23.32±0.11 (0.681±0.002) | **23.78 (0.703)** |
| runtime (sec): |  | 1,190 | 1,584 | **9** | 10 × 9 |

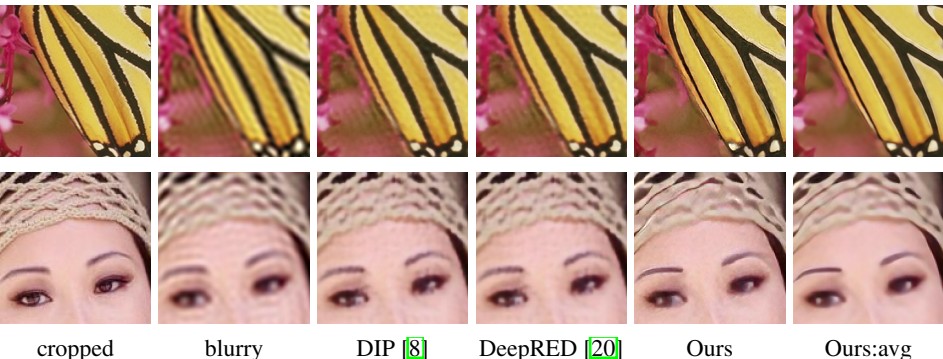

| cropped | blurry | DIP [8] | DeepRED [20] | Ours | Ours:avg |

Figure 4: Deblurring (spectral super-resolution). Measurements correspond to $5\%$ of low frequencies. Original images are from Set5 (butterfly of size $256 \times 256$ and woman of size $224 \times 336$).

**Deblurring (spectral super-resolution).** The applications described above (and in Appendix E) are based on partial measurements in the pixel domain. Here, we consider a blurring operator that retains a set of low-frequency coefficient in the Fourier domain, discarding the rest. This is equivalent to convolving the image with a sinc kernel. In this case, $M$ consists of the preserved low-frequency columns of the discrete Fourier transform, and $MM^Tx$ is the blurred version of $x$. Example images are shown in Fig. 4 and numerical comparisons are shown in Table 2. Our method produces strong results, both perceptually and numerically.

**Compressive sensing.** Compressive sensing [56, 57] aims to recover signals from a small number of (typically, random) linear measurements. In brief, one acquires measurements with a sensing matrix containing a set of $n << N$ random orthogonal axes, and solves the inverse problem by assuming a sparse prior. Photographic images are not truly sparse in any fixed linear basis, but they can be reasonably approximated by low-dimensional subsets of Fourier or wavelet basis functions, and compressive sensing results are typically demonstrated using one of these. The manifold prior embedded within our CNN denoiser corresponds to a nonlinear form of sparsity, and our stochastic coarse-to-fine ascent algorithm can be used to recover an image from the measured linear projections onto the random basis. We compare to four other methods. TVAL3 [53] is an optimization algorithm using total variation regularization, ISTA-Net [54] is a block-based supervised CNN method trained to reconstruct images from measurements obtained from a single pre-specified measurement matrix. BNN [55] is an unsupervised Bayesian method for solving compressive sensing problems. DIP [8] was previously described. Fig. 5 shows results for two example images, and Table 3 summarizes numerical performance. All values are taken from [55] except for ISTA-Net which were obtained by running the open-source code. Our method generally outperforms all other methods, even those that are specialized for Compressive Sensing (ISTA-NET, BNN).

Table 2: Spectral super-resolution (deblurring) performance over Set5, in YCbCr-PSNR (SSIM).

| Ratio | $MM^Tx$ | DIP [8] | DeepRED [20] | Ours±std | Ours:avg |
|---|---|---|---|---|---|
| 10% | 30.2 (0.91) | 32.54 (0.93) | 32.63 (0.93) | 31.82±0.08 (0.93±0.001) | **32.78 (0.94)** |
| 5% | 27.77 (0.85) | 29.88 (0.89) | 29.91 (0.89) | 29.22±0.14 (0.89±0.002) | **30.07 (0.90)** |

Table 3: Compressive sensing performance over Set68 [51]. Values indicate PSNR (SSIM).

| Ratio | TVAL3 [53] | ISTA-Net [54] | DIP [8] | BNN [55] | Ours±std | Ours:avg |
|-------|-----------|---------------|---------|----------|----------|----------|
| 25% | 26.48 (0.77) | 29.07 (0.84) | 27.78 (0.80) | 28.63 (0.84) | 29.16±0.033 (0.88±0.001) | **29.74(0.89)** |
| 10% | 22.49 (0.58) | 25.23 (0.69) | 24.82 (0.69) | 25.24 (0.71) | 25.47±0.03 (0.78±0.001) | **25.84 (0.80)** |
| 4% | 19.10 (0.42) | 22.02 (0.54) | 22.51 (0.58) | **22.52** (0.58) | 22.07±0.05 (0.68±0.002) | 22.29 (**0.69**) |

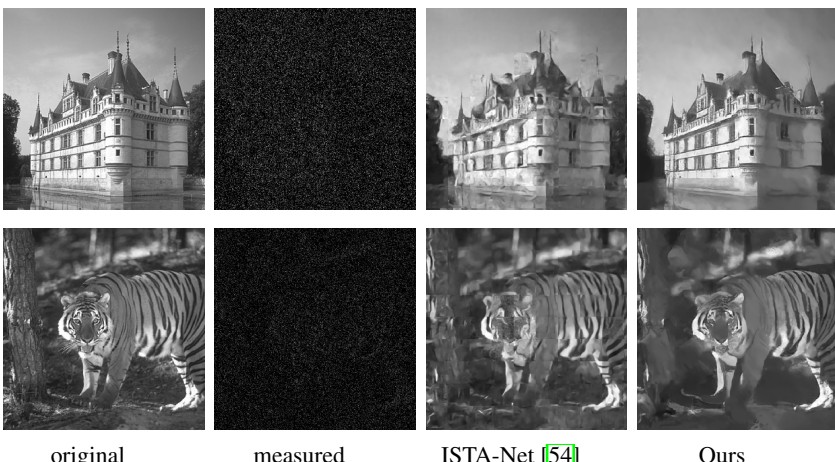

original          measured          ISTA-Net [54]          Ours

Figure 5: Compressive sensing. Measurement matrix $M$ contains random orthogonal unit vectors, whose number is equal to $10\%$ of the number of image pixels ($300 \times 300$). Third column: images recovered using ISTA-Net (trained with supervision for a particular measurement matrix). Fourth column: Our method, which again is seen to exhibit fewer artifacts.

## 4 Related work

Our method is conceptually similar to P&P[14] in that it uses a denoiser to solve linear inverse problems. But it differs in a number of important ways: (1) *direct relationship between prior and denoiser mapping.* The rationale for using a denoiser as a regularizer in P&P frameworks arises from interpreting the proximal operator of the regularizer as a MAP solution of a denoising problem, providing an indirect connection to prior; We derive our method from Miyasawa's relationship between an MMSE denoiser and the prior, which is exact and explicit, and makes the algorithm interpretable. This connection has also been made in [23, 24]. (2) *sampling vs. MAP estimation.* We obtain a high probability image from the implicit prior that is consistent with a linear constraint. Our solution is stochastic, and does not minimize MSE, but has high perceptual quality. RED and other P&P methods are derived as MAP solutions, and although this is not equivalent to minimizing MSE (maximizing PSNR), the results generally have better PSNR than our sampling results, but are visually more blurred (see Figs. 3 and 4); (3) *automatic step-size selection and convergence.* P&P, which uses ADMM for optimization, relies on hyper-parameter selection that are critical for convergence (as discussed extensively in [15]). Our algorithm adjusts step-size automatically using the blind denoiser, with only two primary hyper-parameters ($h_0$ and $\beta$), and convergence is robust to choices of these (Fig. 1).

Our method is also closely related to recent work that uses Score Matching [10] to draw samples from an implicit prior. This line of work is rooted in the connection between Denoising Autoencoders and Score Matching, first described in [32]. Denoising autoencoders learn the gradient of log density, and their connection to an underlying data manifold has been explored in [25, 43, 31]. More recently, Refs. [44, 26] trained a neural network to directly estimate energy (the negative log prior) and interpreted its gradient as the Score of the data. Finally, in a breakthrough paper that partially inspired our approach, Song and Ermon [29] trained a sequence of denoisers with decreasing levels of noise and used Langevin dynamics to sample from the underlying probability distributions in succession. Our work differs from these in several important ways: (1) *Direct derivation.* Our method is based on Miyasawa's explicit relationship between the denoiser mapping and implicit density [9], which can

be proven with a few lines of math (Section 1.3). The relationship of this result to Score Matching [11, 12], and the relationship between Score Matching and Denoising Autoencoders [32] are both significantly more nuanced and complex. (2) *Automatic stepsize adjustment and efficiency.* The algorithm presented in [29] uses a discrete sequence of denoisers, each trained for a single noise level, and embedding a distribution of noisy images with that specific noise level. Langevin dynamics are used to sample from each of these distributions in succession. Langevin sampling guarantees (asymptotically) convergence to each successive distribution, although these intermediate samples are used only for initialization of the subsequent stage. In addition to the Langevin sampling itself, the method requires choices of the schedule of noise levels (standard deviations, and number of iterations used at each level).

In contrast, our formulation relies on a single universal blind denoiser which implicitly embeds an infinite family of distributions of noisy images (corresponding to a continuous range of noise levels). The gradient provided by the denoiser is not used for sampling, but for coarse-to-fine ascent of the noisy distribution, removing a fraction of the noise at each step. The universal denoiser adapts to this updated implicit noisy density, both in terms of gradient direction and magnitude. We use the denoiser magnitude as an estimate of the implicit noise level 6, and this is used to control the amplitude of injected noise. Note that this differs markedly from the control of noise injection in the Langevin method, which is of constant magnitude in each stage. Under assumptions about the denoiser (which hold empirically), our method converges rapidly to a mode of the prior and is robust to parameter choices 1. The continuous maximization of probability results in substantial gains in efficiency. For comparison, Song&Ermon [29] report 1000 iteration to synthesize a $32 \times 32$ image, whereas we synthesize $40 \times 40$ images in roughly 35 iteration (for $\beta = 1$). (3) *Image synthesis vs. linear inverse problems.* Finally, we've focused the use of our method as a universal solver for linear inverse problems, whereas Score Matching methods have generally been focused on unconditional image synthesis.

## 5    Discussion

We've described a framework for using the prior embedded in a denoiser to solve inverse problems. Specifically, we developed a stochastic coarse-to-fine gradient ascent algorithm that uses the denoiser to draw high-probability samples from its implicit prior, and a constrained variant that can be used to solve *any* deterministic linear inverse problem. To demonstrate the generality of our algorithm, we showed solving five inverse problems using the same algorithm and parameters, without any additional training. The derivation is based on a simple and direct expression relating denoising to priors, and a few basic empirical facts about universal CNN denoisers. Finally, we empirically demonstrated its efficiency in Fig. 1 and Table 1. It is worth noting that the assumed use of Gaussian additive noise and MSE objective in training the denoiser is necessary only to justify the use of Miyasawa's expression (Eq. 3), but does not impose any such restrictions on use of the trained denoiser for sampling from the implicit prior or solving inverse problems. Denoisers can be trained supervised on unlimited amounts of unlabeled data, and as such, our method extends the power of supervised learning to a much broader set of problems, with no additional training.

As discussed following Eq. (5), the control of step sizes and noise injection in our algorithm is chosen by assuming that the denoiser residual exactly cancels the noise. We showed empirically (Fig. 1) that this idealization does not disrupt convergence for BF-CNN, but we cannot guarantee this would hold for all other denoisers. More generally, although the method can be used with any universal least-squares denoiser designed or trained to remove Gaussian noise, the complexity of the embedded prior, and thus the quality of our linear inverse results, relies heavily on the expressive power of the denoiser (BF-CNN in our case), as well as the diversity of the training set. Finally, the current algorithm is designed to solve *deterministic linear* inverse problems. We are currently exploring extensions to inverse problems based on stochastic or nonlinear measurements.

## Acknowledgments and Disclosure of Funding

We gratefully acknowledge financial support from the Simons Foundation, the Howard Hughes Medical Institute, and NSF NRT HDR Award 1922658 to the Center for Data Science at NYU. High performance computing resources were provided by the NYU HPC center and the Flatiron Institute of the Simons Foundation.

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
