# A Description of BF-CNN denoiser

Examples in this paper were computed using the publicly-available implementation of the BF-CNN denoiser [42], which is trained to minimize MSE for images corrupted with Gaussian white noise of unknown variance.

**Architecture.** The network contains 20 bias-free convolutional layers, each consisting of $3 \times 3$ filters and 64 channels, batch normalization, and a ReLU. To construct a bias-free network, we remove all sources of additive bias, including the mean parameter of the batch-normalization.

**Training Scheme.** We follow the training procedure described in [42]. The network is trained to denoise images corrupted by i.i.d. Gaussian noise with standard deviations drawn from the range $[0, 0.4]$ (relative to image intensity range $[0, 1]$). The training set consists of overlapping patches of size $40 \times 40$ cropped from the Berkeley Segmentation Dataset [51]. Each original natural image is of size $180 \times 180$. Training is carried out on batches of size 128, for 70 epochs. Additionally, we train two other BF-CNN denoisers on MNIST [52] and color Berkeley segmentation dataset [51].

# B Block diagram of Universal Linear Inverse Solver

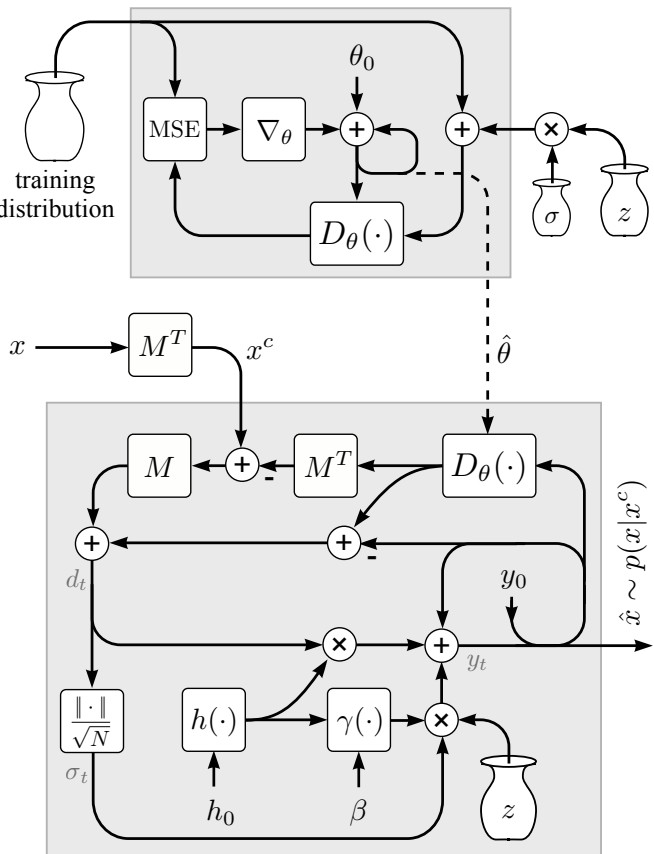

Figure 6: Block diagrams for denoiser training, and Universal Inverse Sampler. **Top:** A parametric blind denoiser, $D_\theta(\cdot)$, is trained to approximate $\hat{x}(y)$ by minimizing mean squared error when removing additive Gaussian white noise ($z$) of varying amplitude ($\sigma$) from images drawn from a training distribution. The trained denoiser parameters, $\hat{\theta}$, constitute an implicit model of this distribution. **Bottom:** The trained denoiser is embedded within an iterative computation to draw samples from this distribution, starting from initial image $y_0$, and conditioned on a low-dimensional linear measurement of a test image: $\hat{x} \sim p(x|x^c)$, where $x^c = M^T x$. If measurement matrix $M$ is empty, the algorithm draws a sample from the unconstrained distribution. Parameter $h_0 \in [0, 1]$ controls the step size, and $\beta \in [0, 1]$ controls the stochasticity (or lack thereof) of the process.

## C Visualization of Universal Inverse Sampler on a 2D manifold prior

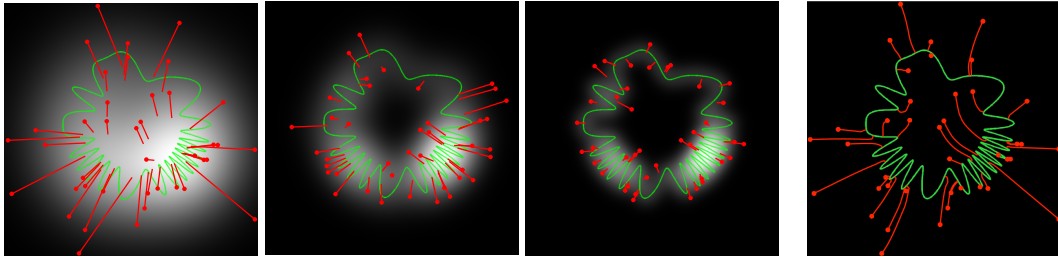

Figure 7: Two-dimensional simulation/visualization of the Universal Inverse Sampler. Forty example signals $x$ are sampled from a uniform prior on a manifold (green curve). First three panels show, for three different levels of noise, the noise-corrupted measurements of the signals (red points), the associated noisy signal distribution $p(y)$ (indicated with underlying grayscale intensities), and the least-squares optimal denoising solution $\hat{x}(y)$ for each (end of red line segments), as defined by Eq. (2), or equivalently, Eq. (3). Right panel shows trajectory of our iterative coarse-to-fine inverse algorithm (Algorithm 2, depicted in Figure 6), starting from the same initial values $y$ (red points) of the first panel. Algorithm parameters were $h_0 = 0.05$ and $\beta = 1$ (i.e., no injected noise). Note that, unlike the single-step least-squares solutions, the iterative trajectories are curved, and always arrive at solutions on the signal manifold.

## D Sampling - more examples

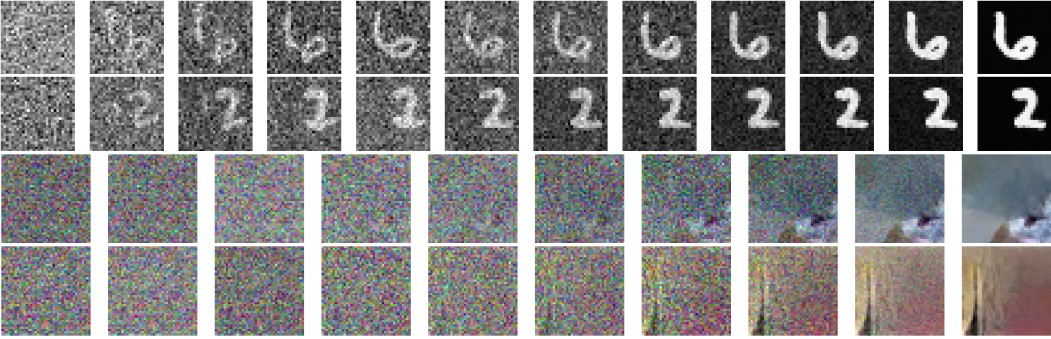

Figure 8: Sampling from the implicit prior embedded in a BF-CNN denoiser trained on MNIST dataset (first two rows) and Berkeley segmentation dataset (second two rows).

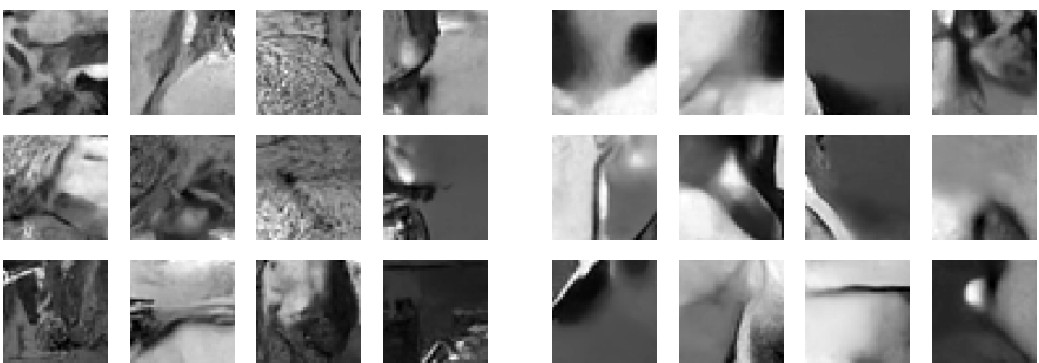

Figure 9: **Left.** Samples drawn with different initializations $y_0$, using a moderate level of injected noise ($\beta = 0.5$). Images contain natural-looking features, with sharp contours, junctions, shading, and in some cases, detailed texture regions. **Right.** Samples drawn with more substantial injected noise ($\beta = 0.1$). The additional noise helps to avoid local maxima, and arrives at images that are smoother and higher probability, but still containing sharp boundaries.

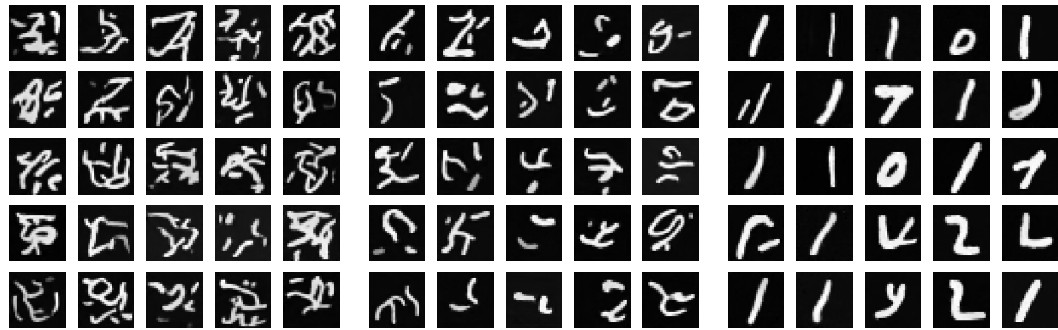

Figure 10: Training BF-CNN on the MNIST dataset of handwritten digits [58] results in a different implicit prior (compare to Figure 9). Each panel shows 16 samples drawn from the implicit prior, with different levels of injected noise (increasing from left to right, $\beta \in \{1.0,\ 0.3,\ 0.01\}$).

## E   Linear inverse problem - more examples

---

**Algorithm 2:** Coarse-to-fine stochastic ascent method for sampling from $p(x|M^T x = x^c)$, based on the residual of a denoiser, $f(y) = \hat{x}(y) - y$. Note: $e$ is an image of ones.

---

parameters: $\sigma_0$, $\sigma_L$, $h_0$, $\beta$, $M$, $x^c$
initialization: t=1; draw $y_0 \sim \mathcal{N}(0.5(I - MM^T)e + Mx^c,\ \sigma_0^2 I)$
**while** $\sigma_{t-1} \geq \sigma_L$ **do**
   $h_t = \frac{h_0 t}{1 + h_0(t-1)}$;
   $d_t = (I - MM^T)f(y_{t-1}) + M(x^c - M^T y_{t-1})$;
   $\sigma_t^2 = \frac{||d_t||^2}{N}$;
   $\gamma_t^2 = \left((1 - \beta h_t)^2 - (1 - h_t)^2\right)\sigma_t^2$;
   Draw $z_t \sim \mathcal{N}(0, I)$;
   $y_t \leftarrow y_{t-1} + h_t d_t + \gamma_t z_t$;
   $t \leftarrow t + 1$
**end**

---

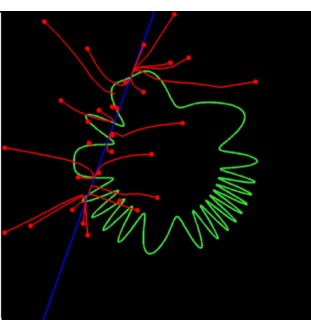

Figure 11: Two-dimensional simulation/visualization of constrained sampling. Only points lying at the intersection of manifold and constraint hyperplane (represented by low-ranked matrix $M$) are valid samples.

**Inpainting.** A simple example of a linear inverse problem involves restoring a block of missing pixels, conditioned on the surrounding content. Here, the columns of the measurement matrix $M$ are a subset of the identity matrix, corresponding to the measured (outer) pixel locations. We choose a missing block of size $30 \times 30$ pixels, which is less than the size of the receptive field of the BF-CNN network ($40 \times 40$), the largest extent over which this denoiser can be expected to directly capture joint statistical relationships. Figure 12 shows examples.

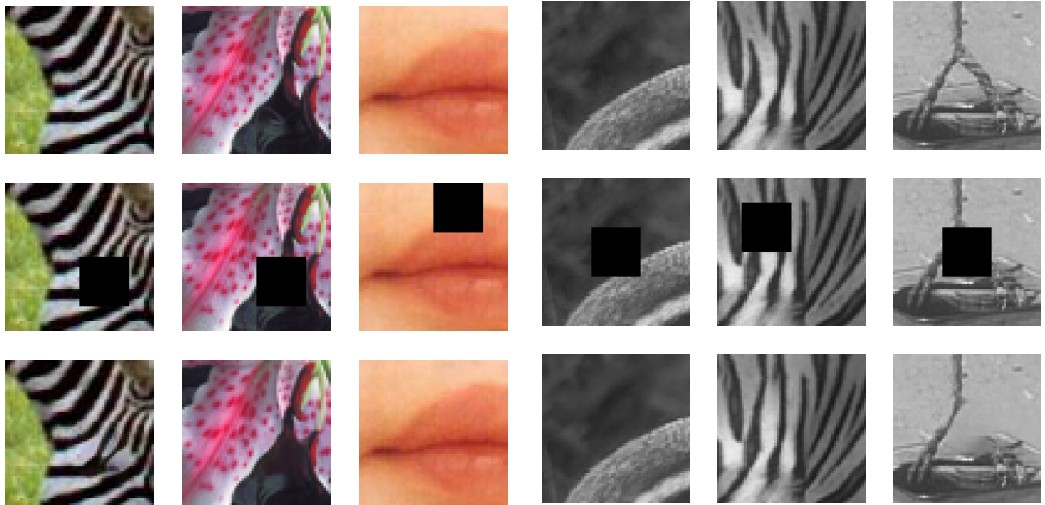

Figure 12: Inpainting examples. Top row: original images ($x$). Middle: Images corrupted with blanked region ($MM^T x$). Bottom: Images restored using our algorithm.

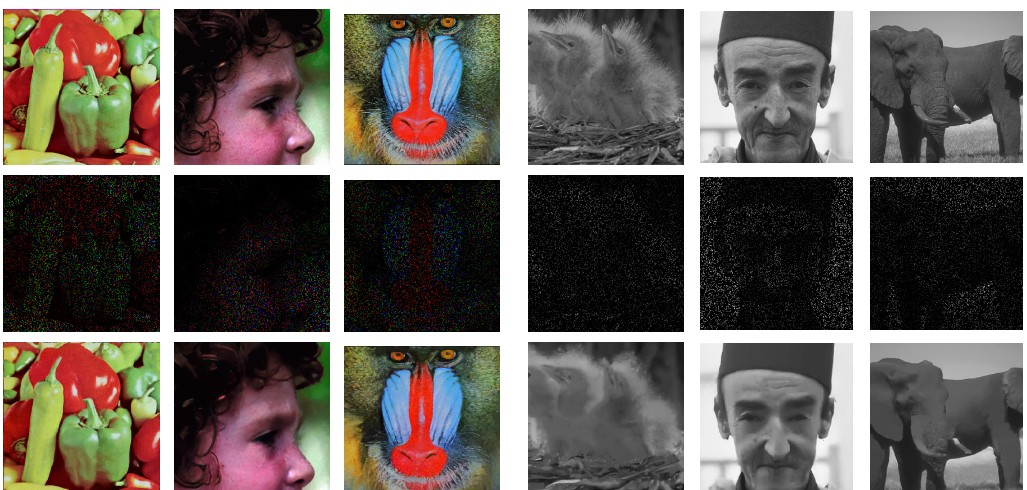

Figure 13: Recovery of randomly selected missing pixels. $10\%$ of dimensions retained.

**Random missing pixels.** Consider a measurement process that discards a random subset of pixels. $M$ is a low rank matrix whose columns consist of a subset of the identity matrix corresponding to the randomly chosen set of preserved pixels. Figure 13 shows examples with $10\%$ of pixels retained. Despite the significant number of missing pixels, the recovered images are remarkably similar to the originals.

Table 4: Run time, in seconds, and (number of iterations until convergence) for different applications, averaged over images in Set12, on an NVIDIA RTX 8000 GPU.

| Inpainting $30 \times 30$ | Missing pixels 10% | SISR 4:1 | Deblurring 10% | CS 10% |
|---|---|---|---|---|
| 15.7 (647) | 7.8 (313) | 7.6 (301) | 8.4 (332) | 32.5 (317) |

The cost of running the algorithm can be divided into two parts: number of iterations, and cost per iteration. Number of iterations for all applications except for inpainting is in the same order. Solving inpainting with this algorithm requires information to spread in from the borders of the missing block, which in turn requires more iterations. Cost of each iteration, aside from the cost of denoiser's forward pass, comes from $MM^Tx$ operation. For all applications, except for CS, dot products can be reduced to more efficient operations such as Hadamard products or filtering. In CS, $M$ contains random vectors, so measurements require full matrix multiplication. Consequently, CS takes longer per iteration.