# OpenReview forum: "Stochastic Solutions for Linear Inverse Problems using the Prior Implicit in a Denoiser"
_NeurIPS.cc/2021/Conference — NeurIPS 2021 Poster_

### Official Review · Reviewer_9hAm · 2021-07-07

**Rating:** 7
**Confidence:** 4

**Summary:**

The paper derives an algorithm for solving linear inverse problems using the prior that is implicit in an end-to-end denoiser in the form of a CNN.

Consider a denoising algorithm in form of a CNN that takes a noisy observation $y = x+z$ as input and yields an estimate of the signal $x$. CNNs trained end-to-end yield state-of-the-art denoising performance. After training, the image/signal prior is implicit in the CNN. The paper notes that the least-squares-estimate of the signal given a prior $p(x)$ is

$$
\hat x(y) = y + \sigma^2 \nabla_y \log p(y),
$$

where $p(y)$ is the noisy observation density. Based on this relation, the paper proposes a method to obtain a high-probability image (sample) from the prior implicit in a denoiser. The idea is as follows: We can denoise a given image by evaluating $\hat x(y)$, and by computing the residual $f(y) = \hat x(y) - y$ we obtain an image that is proportional to $\nabla_y \log p(y)$. Based on this insight, the paper proposes to sample from the prior implicit in a denoiser by running stochastic gradient ascent through iterating $y_{t+1} = y_t + h_t f(y_t) + \gamma_t z_t$, where $z_t$ is Gaussian noise, and $h_t$ and $\gamma_t$ are tuning parameters. The paper demonstrates (Fig. 2) that this method can yield realistic samples of low-dimensional images.

Next, the paper then modifies this approach to solve linear inverse problems by obtaining a high-probability sample based on partial linear measurements of a signal/image. The paper evaluates the method on standard inverse problems, including super-resolution, deblurring, and compressive sensing. The paper shows that the method works well, both in numbers and by providing example images.

**Ethical Concerns:**

None.

**Limitations And Societal Impact:**

Overall the paper adequately discusses limitations. One interesting comparison to add is U-net-based reconstruction in Table 3 (so U-net applied to M^Ty, where y is the measurement), along with corresponding computational costs. I expect that this will show that U-net-based reconstruction is at least on par in terms of image quality, while being faster.

**Main Review:**

The paper introduces an original way to solve linear inverse problems and demonstrates with simulations that it works well, both computationally and in terms of imaging quality. The idea is well explained, the paper is well written, and the experiments support the claims. I would be happy to see the paper at the conference.

Here are a few comments/experiments that would make the paper even stronger:
- The paper uses as an denoiser the bias-free CNN from [36], and writes it obtains similar results with other-bias free denoisers. I would be curious to also see results for other standard CNN-denoisers, in particular for U-net, because U-net or similar encoder-decoder architectures work very well for denoising and other inverse problem. If the proposed approach does not work well with U-net, that wouldn't make the paper any less interesting of course. I would also be interesting to show results for standard CNN denoisers that are not bias-free, and potentially add a brief discussion whether bias-freeness ist important or not for this method to work.
- For the numbers in Table 1 ``ours - avg'', it would be great to also give confidence intervals or a standard deviation, to get intuition on the variation across samples.
- Another interesting baseline that is also easy to train for table 3 is a U-net trained end-to-end for compressive sensing. Runtimes for CS would also be great to add.
- Can the method also give realistic samples of high-resolution images? I didn't understand from the paper what the resolution of the images in figure 4 or 5 is.


**Time Spent Reviewing:**

4

---

> ### Author Response · Authors · 2021-08-10
> **Response to reviewer 9hAm**
>
> Thank you for your constructive comments and suggestions.
>
> *Bias-free vs. Standard CNN denoiser*: Thank you for pointing this out. We agree that the paper would benefit from a short discussion on the role of bias-freeness. Our algorithm relies on the denoiser being blind and universal, allowing it to (implicitly) estimate noise level to control the step size. Mohan et al [Ref 36] showed that the bias terms of universal standard CNN denoisers are driven toward zero when trained over a wide range of noise levels, and that this holds for different architectures. So, in practice, any universal denoiser used in our algorithm seems likely to have small net bias, even with a standard architecture. The advantage of enforcing bias-freeness on the architecture is that the learned denoiser is guaranteed to be universal even if trained on a narrow range of noise.
>
> *Other denoiser architectures*: we have tested all of the architectures discussed in (Mohan et al, 2020), including Recurrent-CNN, Dense-Net, and U-Net. Performance was nearly identical across all the linear inverse applications. But your suggestion has reminded us that the specific U-Net denoiser we used was a truncated one provided by the online code of Mohan et al. So we will revisit this using the original U-Net architecture, which may lead to improvements in speed or quality  - thanks for the suggestion!
>
> *Standard deviation of PSNR results*: Thank you for mentioning this - we agree it would be useful to include that for single sample results (columns labeled as “ours”), given the stochastic nature of our algorithm. Note that the columns labeled “ours - ave”  show PSNR of the average of many samples (as opposed to the average of many single sample’s PSNRs), so assuming we’ve averaged over enough samples these values should have converged to a stable value.
>
> *Compressive sensing*: Supervised solutions for compressive sensing are generally trained on a single (randomly chosen) measurement matrix. This is a strong restriction compared to the general case (an inverse method that can operate on any measurement matrix), but it can produce high-quality and efficient solutions.  As an example of this treatment, we have shown ISTA-Net results in the paper.  But we’re not sure what is meant by “an end-to-end solution that operates on the measurements”.  In particular, since the measurements are not convolutional, they would seem unsuitable for input to a U-net. Regarding end-to-end training, we do have an interest in trying to optimize the measurement matrix. This is a difficult problem (since the decoder is an iterative stochastic procedure).  But since the prior implicit in the denoiser is not a union of subspaces (as assumed by the original Compressive Sensing methods), there may be measurements that are significantly more effective than random projections in capturing image information.
>
> *Compressive sensing runtimes*: Run times for different applications (including compressive sensing) are shown in Appendix Table 4. Our average runtime for CS (10% dimensionality) is 32.5 seconds - note that this is slower than the other applications, simply because the measurements require multiplication by a dense matrix rather than a diagonal or convolutional matrix. For the comparative results in Table 3, apart from ISTA-Net, we have used the reported performance values  and run times from the original papers. With that caveat in mind, it is still worth noting that the average runtime of the second best unsupervised method (BNN) is 1680 seconds (or 27 minutes) on TITAN RTX GPU as reported in the cited paper. Runtime for ISTA-Net is less than one second, but as mentioned above,  it is restricted to a single measurement matrix (on which it is trained) while our method can operate on any measurement matrix. We will include additional comparative runtimes in the final paper.
>
> *Image resolutions*: Thank you for pointing this out. In general, the algorithm can be applied to images of any resolution. In Figure 5, both images are 300x300. In figure 4, the full “butterfly” image is 256x256 and “woman” is 224x336, but we only show cropped portions to allow the reader to see qualitative differences. We will add a column to the figure to show the full original and the full recovered images. The values reported in the paper’s tables are from popular image sets (Set5, Set14, Set68), and these images are of varying resolution. We will specify image resolution in the captions of the final version.

---

### Official Review · Reviewer_TGbx · 2021-07-15

**Rating:** 6
**Confidence:** 3

**Summary:**

This paper proposed a score-based method that solves deterministic linear inverse problems by drawing samples from the denoising priors. By iteratively maximizing the score $\textsf{log}p(y)$ using Langevin process, high-probability samples can be drawn from the distribution $p(x)$ embedded in the denoiser. However, in order to perform this, $\nabla_y\textsf{log}p(y)$ needs to be effectively computed. By leveraging the classic Miyasawa’s equation for MSE denoisers, the authors are able to use a residual CNN denoiser trained for minimizing MSE to approximate the term $\textsf{CNN}(y) = \hat{x}(y)-y = \nabla_y\textsf{log}p(y)$.
The authors also extend their method to linear deterministic linear inverse problems by expressing $\textsf{log}p(y)$ into two orthogonal components accounting for noise density and constraint, respectively.

The method is related to the recent Plug-and-play (RED) and Regularization-by-denoising (RED) frameworks which take an optimization-centric view of using denoisers. However, it differs from the former in the sense that it explicitly relates the denoiser with the prior distribution. In the experiments, one self-supervised neural network and one RED algorithm are used as baselines, achieving inferior performance compared with the proposed method in single image super-resolution, denoising, deblurring, and compressive sensing.

**Limitations And Societal Impact:**

1. While reading the manuscript, I encountered some confusions that need authors' explanation: 1) Throughout the paper, it seems to me that the final output is $\hat{x}(y)$. However, the algorithm summarized in Alg. 1 seeks to update $y$, which I assume to be related to the noisy observation. Is there a hidden step $\hat{x}(y)=f(y)+y$ for dumping the final output? or am I misinterpreting the variables? 2) Sampling $y$ seems quite unusual if my interpretation of $y_t$ is correct. In [1], I saw their formulations are for $x$. 3) I did not quite see why from the equation above eq 7 we can have eq 7. Could the authors provide some intuitive explanations? Derivations are also welcome.

2. A discussion between your method and [1] is needed.

3. Is it possible to construct a simple toy problem such that the oracle MMSE denoiser is reachable. In this way, we can compare CNN with this oracle denoiser, which may bridge the gap between theory and experiment. Can the experimental setup (Compressive Sensing with Bernoulli-Gaussian distribution) in [2] directly reusable to yours?

4. In experiments, deepRED is treated as the PnP baseline. Although RED and PnP are closely related, the exact equivalence between the two is still an open question. I suggest including one exact PnP baseline (PnP-ISTA or PnP-ADMM) in the experiments.

5. Minor: In line 245 Section 4, the sentence implies that PnP is coupled with ADMM. However, this is a long-standing misunderstanding of the PnP framework. Many works have shown that PnP is compatible with gradient-based [3], primal-dual [4], and message passing algorithms [5]. 2)


[3] Kamilov et al. A Plug-and-Play Priors Approach for Solving Nonlinear Imaging Inverse Problems\
[4] Ono et al. Primal-Dual Plug-and-Play Image Restoration\
[5] Metzler et al. Learned D-AMP: Principled Neural Network-Based Compressive Image Recovery




**Main Review:**

**Originality:** This work is novel in relating the MSE denoisers to priors in an explicit way by using the classic result. The difference between PnP/RED and the proposed method is clearly stated. However, I found a concurrent paper [1] that seems to propose a similar method. I believe a discussion between your and their methods can avoid future confusion. Moreover, investigations on the relationship between denoisers and explicit regularizers have also been conducted for MMSE denoisers within the PnP framework [2].

[1] Laumont et al. 'Bayesian imaging using Plug & Play priors: when Langevin meets Tweedie\
[2] Xu. et al. Provable Convergence of Plug-and-Play Priors With MMSE Denoisers

**Quality:** This paper conducts adequate experiments to support their claims. The results are technically sound and convincible. For questions please see my comments below. Also, the limitations of the proposed method are well-discussed.

**Clarity:** Since I am not very familiar with the score-match field, the paper is a little bit difficult for me to follow. A supplement with some introductory details (e.g. derivation of Langevin process) will be welcome for readers like me.

**Significance:** This paper provides interesting insights from a Bayesian perspective, which can be useful for the theoretical studies of PnP/RED.



**Time Spent Reviewing:**

2 hours for writing response. 3 hours for reading and evaluation

---

> ### Author Response · Authors · 2021-08-10
> **Response to reviewer TGbx**
>
> Thank you for your constructive comments and suggestions. In particular, thanks for pointing us to the recent related publications [1] and [2], which we’d not seen.  We will include them in the discussion of our revised paper.
>
> *Variable notation*: We apologize for the confusion. We’ve used the same variables in describing the denoiser, and in describing our iterative sampling/inverse procedure.  Specifically, $\hat{x}(y)$ represents the denoiser solution (mean of the posterior), and through Miyasawa’s formula, is equal to the sum of the noisy image y and the $\nabla\log p(y)$. Thus, the denoiser residual, $f(y)  = y- \hat{x}(y)$ can be used to compute the $\nabla\log p(y)$.  For the sampling/inverse procedure, we start from a noise image (which we label $y_0$), and we iteratively update $y_t$ by adding a multiple of $f(y_t)$, and injecting additional noise $(z_t)$.  The scaling of both terms is controlled by the denoiser.
>
> *Eq 7 clarification*: In the paragraph leading to Eq 7, we provided an intuition for the derivation, but thanks to your comment, we realized that is too terse. The second term is the gradient of the log of a Gaussian which is constrained to lie in the measurement subspace (column space of M). Taking the log and gradient of that Gaussian reduces it to the second term in Eq 7 . The first term is the gradient of log of an unknown distribution in the space orthogonal to the measurement space. That is computed by taking the denoiser output $f(y)$,  (i.e. the estimated gradient log in the entire space) and projecting it onto the orthogonal complement of the measurement space. Specifically, this is achieved by multiplying $f(y)$ with $(I - MM^T)$ which is the projection matrix onto the orthogonal complement of M. Hope this helps - we will rework and elaborate the text description around Eq 7.
>
> *Simple toy problem*: Thanks for this suggestion. It is challenging to find an example with nontrivial prior where the MMSE solution can be computed exactly, but we have tested our algorithm on simple low-dimensional manifolds (eg, see Fig 7).  We’ve also looked at simple low-dimensional priors in high dimensions (e.g., all translations of a sinusoidal grating image), verifying that the algorithm converges to samples on the manifold. We can incorporate these results into the Appendix of the revised paper.
>
> *Additional PnP baselines*: We chose RED as an example PnP because of its popularity and its successful performance. However, due to your comment, we have become aware that some authors have questioned whether RED should be classified as a PnP method. In our final version we will add another PnP baseline to our comparative tables.
>
> *PnP not coupled with ADMM*: Thank you for pointing this out. We are happy to incorporate this more nuanced description into our text, and will cite the publications you mention.
>
> *Regarding “Clarity”*: We would like to emphasize that although our update equations may appear similar, our algorithm does not use Langevin dynamics (or any other MCMC sampling method).  The algorithm is directly derived as a stochastic ascent procedure, based on the Miyasawa equation (Eq 3). Although it is a bit terse, we tried to write a self-contained derivation by introducing the necessary background in the Introduction section leading to Eq 4 and eventually to the full algorithm in section 3.

---

> > ### Comment · Reviewer_TGbx · 2021-08-12
> > **1-D toy problem**
> >
> > Thank you for the response. In fact, the authors of [2] seem to construct a toy problem where the MSE denoiser can be constructed (as far as they show in the figures). Would their setup be directly reusable?

---

> > > ### Author Response · Authors · 2021-08-14
> > > **Toy problem**
> > >
> > > Thanks for prompting us to think a bit more about the idea of testing our inverse algorithm on a toy problem where the prior is known.  We’ve now taken a more careful look at Ref [2], which proves that an MMSE denoiser can be used in an ISTA implementation of PnP (via Tweedie’s formula, for which we have been referencing Miyasawa’s 1961 paper).  They then demonstrate this empirically on a simple example using a hierarchical Bernoulli-Gaussian prior, and also show that these outperform simple LASSO (but are inferior to GAMP).
> > >
> > > It is hard to predict how our iterative inverse algorithm would fare on this particular test case.  Specifically, the Bernoulli-Gaussian prior is sparse, but not restricted to a manifold, and so outside the class of priors for which we’ve designed our method.  Perhaps more importantly, our method is built around a *universal* MMSE denoiser (one that can operate on images corrupted by noise of unspecified amplitude, $\sigma$), and relies on this property  to automatically control step sizes.  In order to use it on the content of [2], we’d need to incorporate an estimate of $\sigma$ - for example, we could compute the MLE for $\sigma$ at each step of the iteration, and plug this in to compute the MMSE estimator.  To be clear, this is exactly how the simulations shown in the rightmost panel of Fig 7 and in Fig 11 were done - we assumed a uniform prior on a sparse manifold  (green curve), and on each iteration, computed the MLE for $\sigma$ (numerically), plugged this in to compute the MMSE estimate (again, numerically), and then used the residual $f(y_t)$ in our algorithm.  Your question leads us to realize that we should provide quantitative analysis of the performance (e.g., convergence, and distance from manifold at converged solution) compared to ISTA-MMSE, on both the manifold example of Figs 7/11, and on the Bernoulli-Gaussian example.   We intend to do this for the revised paper.

---

> > > > ### Comment · Reviewer_TGbx · 2021-08-22
> > > > **Looking forward to the results**
> > > >
> > > > Thanks for taking a closer look at [2]. Looking forward to your new comparisons in the revision.

---

### Official Review · Reviewer_tt2X · 2021-07-16

**Rating:** 7
**Confidence:** 3

**Summary:**

Classical result by Miyasawa [25]  shows that the gradient of the log of the noisy signal density  can be expressed in terms of least-squares optimal denoisers for additive Gaussian noise.  Using this result, the authors develop an algorithm which uses  stochastic gradient ascent  to sample from the implicit prior of the denoiser, where the gradients  are computed using the residual of a trained bias-free universal CNN denoiser.  The algorithm includes noise injection which allows stochastic sampling of the manifold. They subsequently modify this algorithm to account for constraints from noise-less linear measurement model to solve linear inverse problems. The authors perform different linear image reconstruction tasks including spatial super-resolution, deblurring, compressive sensing, inpainting and reconstruction from random missing pixels. Experimentally, they demonstrate faster and visually better reconstructions as compared to the considered baselines.

**Limitations And Societal Impact:**

The limitations are adequately discussed.

**Main Review:**

The paper is nicely motivated, well-written and clear. The overall topic is highly relevant and the numerical results are encouraging. Yet, despite the detailed explanation given in lines 255 to 272, the difference to [20] did not become perfectly clear to me from an algorithmic perspective and thus raises questions about the originality of the research in my oppinion. The authors state that their method is not based on Langevin dynamics. The update equation for y in eq.(4) is, however, identical to the update equation of x in [20] except for the way the scaling factors / step sizes are computed. Both approaches update the variable with the help of (differently weighted) combinations of the denoiser residual and random noise. Thus, I believe it is crucial to discuss the difference of the proposed approach with respect to Langevin dynamics in more detail. In particular, the Langevin dynamics of [20] could be used / compared to in the numerical results. As the paper is validated empirically (with no convergence proof of the proposed scheme), I believe it is crucial to describe why the two approaches are different despite their very similar appearance of the algorithm.

To better describe the relation of the proposed work in the overall context of reserach on using denoising priors, I am attaching some related works below, which could be of interest. (Some of them are preprints only and of course to not have to be considered, yet could be of general interest to the authors).

Typos:
 - line 118: maniofld >> manifold
 - line 126- 127: residual of the denoising function proportional to the gradient of p(y) >>  residual of the denoising function proportional to the gradient of log p(y)



[a] Guo, Bichuan, Yuxing Han, and Jiangtao Wen. "Agem: Solving linear inverse problems via deep priors and sampling." Advances in Neural Information Processing Systems 32 (2019).

[b] Ramzi, Zaccharie, Benjamin Rémy, Francois Lanusse, Jean-Luc Starck, and Philippe Ciuciu. "Denoising Score-Matching for Uncertainty Quantification in Inverse Problems." In NeurIPS 2020-34th Conference on Neural Information Processing Systems/Workshop on Deep Learning and Inverse Problems. 2020

[c] Regev Cohen, Michael Elad, and Peyman Milanfar. "Regularization by denoising via fixed-point projection (red-pro)." (2020)

[d] Kawar, Bahjat, Gregory Vaksman, and Michael Elad. "SNIPS: Solving Noisy Inverse Problems Stochastically." arXiv preprint arXiv:2105.14951 (2021)

[e] Saharia, Chitwan, Jonathan Ho, William Chan, Tim Salimans, David J. Fleet, and Mohammad Norouzi. "Image super-resolution via iterative refinement." arXiv preprint arXiv:2104.07636 (2021).

[f] Laumont, Rémi, Valentin De Bortoli, Andrés Almansa, Julie Delon, Alain Durmus, and Marcelo Pereyra. "Bayesian imaging using Plug & Play priors: when Langevin meets Tweedie." arXiv preprint arXiv:2103.04715 (2021)

-----------------------------
I would like to thank the authors for their explanation which helped me understand the difference to Langevin dynamics better. Due to the large algorithmic similarity to Langevin dynamics I would have still appreciated a comparison to (or at least a small test on) using an algorithm like [20] in the same fashion the authors are solving inverse problems based on the idea of algorithm 1. This would contribute to my understanding whether it is the use of stochasticity or the specific way it is introduced (i.e. controlling the amount of noise via the magnitude of the residual predicted by the denoiser) that leads to the superior results. Although it at least seems like [20] could easily be adapted to solve inverse problems the same way as proposed in this paper, I understand that [20] addressed generative tasks only (counting inpainting as a generative task). I will therefore raise my score to 7.

**Time Spent Reviewing:**

2

---

> ### Author Response · Authors · 2021-08-10
> **Response to reviewer tt2X**
>
> Thank you for your constructive comments and suggestions.
>
> From an algorithmic perspective, you are right: the update line in our algorithm is of similar form to the update step in [20] (it also is of the similar form as that of [38], and several others).  However, in addition to differences in conceptualization and derivation, the primary algorithmic difference from our work lies in the schedule used to adjust step sizes. The algorithm presented in [20] uses a discrete sequence of denoisers, each of which is trained for a single noise level, and embeds a distribution of noisy images with that specific noise level. Langevin dynamics are used to iteratively sample from each of these distributions in succession (the outer loop), with each stage initialized from the sample generated by the previous stage. This procedure is repeated until the noise is sufficiently small (annealed Langevin dynamics). The convergence to every intermediate distribution is inherited from the Langevin sampling, which (under some assumptions) converges to a stationary distribution. In addition to the Langevin sampling itself, the method requires choices regarding the schedule of noise levels (both the set of standard deviations, and the number of iterations used at each level).
>
> In our formulation, we conceptualize sampling from the prior embedded in a denoiser as a coarse-to-fine ascent procedure. We use a single universal blind denoiser which implicitly embeds an infinite family of distributions of noisy images (corresponding to a continuous range of noise levels). Starting from a random initial point, the denoiser is used to ascend the gradient of the noisy distribution, removing a fraction of the noise at each step. The universal denoiser adapts to this updated implicit noisy density, both in terms of gradient direction and magnitude. We use the denoiser magnitude as an estimate of the implicit noise level (Eq 6), and this is used to control the amplitude of injected noise. Note that this is quite different from the control of noise injection in the Langevin method, which is of constant magnitude in each stage. Under assumptions about the denoiser (which hold empirically), our method converges rapidly and is robust to parameter choices (Fig 1). We will incorporate some of this explanation into the revised paper.
>
> We also note that the numerical results in [20] are primarily aimed at image synthesis (i.e., sampling from the prior), whereas our paper is focused primarily on solving linear inverse problems. The only linear inverse problem discussed in [20] is inpainting, for which numerical PSNR comparisons are not so meaningful.
>
> Finally, thank you for the useful list of references. We were aware of some, but not all, and will include them in the revision.

---

### Official Review · Reviewer_N12S · 2021-07-20

**Rating:** 6
**Confidence:** 5

**Summary:**

The authors proposed a stochastic coarse-to-fine gradient ascent method for drawing high-probability samples from the implicit prior to denoise the noisy image and generalizes it to more broad linear inverse problems.

**Limitations And Societal Impact:**

Paper organization:
The proposed method and experimental results are mixed in Section 3, which does not help readers understand the proposed solution better. It is more readable to separate the methodology and experiments into two sections and clearly present the methodology contribution.

There is only a subsection in Section 2. Also, in Section 2.1, the title Image synthesis is not logically connected with the denoiser, which should be explicitly explained. Also, this section is about experiments and should be merged with Section 3.2.

Technical issues
Line 173-174: This part is not clear since \bar{M} is not clearly defined. How y is decomposed as (y^c, y^u) should be explicitly explained using the property of orthogonal complement. Also, why y^c can be omitted conditional on x^c in p(y^u|y^c, x^c)?
Last but not least, the dimension of y and x is very likely not the same in super-resolution, inpainting, compressive sensing, etc. Let’s say, x in R^n and y in R^m, thus m<n. If M^T can be left multiplied to x, how can it be also left multiplied to y as well?

Line 74: The authors’ claim that (2) is least square estimation is not correct. The conditional mean of the posterior is the MMSE estimator.

In Fig 6, the notation of denoiser D is not consistent with the text and Algo 1, which can easily confuse readers.

Note that the proposed method requires Monte Carlo sampling, which might be quite time-consuming. The computation complexity needs to be analyzed and in experiments, the running time for different methods should be compared.

Minor issues:

Line 78: Solves—> solves it.

In Fig 3, the reference for DeepRED is not correct.



**Main Review:**

The idea is quite interesting, i.e., building a plug-and-play denoiser to inverse problems.
However, the proposed approach is not very clearly explained. There are many incorrect or unclear claims, which are detailed in the below box.
The paper is also not very well organized, e.g., methodology and experiments are mixed in one section. The authors' contribution in methodology is not clearly separated from existing works.

**Time Spent Reviewing:**

7

---

> ### Author Response · Authors · 2021-08-10
> **Response to reviewer N12S**
>
> Thank you for the helpful feedback.  Our specific responses are below.
>
> Organization: We apologize for any confusion arising from the organization of the paper. We thought it would be helpful to readers to first see the derivation of the simpler (unconstrained synthesis) algorithm along with some examples.  Those are presented in Section 2 (which we agree should be partitioned into 2.1 and 2.2). This is not the primary contribution of the paper, but is meant to demonstrate basic properties of the algorithm (convergence), properties of the prior visible in generated images (sharp edges, smooth regions, etc) and the effects of the hyper parameter (injected noise). Section 3 then generalizes the algorithm to include constraints from linear measurements, and shows its application to 5 different inverse problems (this is the main contribution). We will do our best to improve the transitions and section labeling to make the structure easier to follow.
>
> Technical questions: We agree that our derivation is a bit terse, and we will elaborate and clarify some of the details in the finalized paper.  Specifically:
>
> (1) the vector $y$ is projected onto two complementary subspaces (lines 173, 174) - the matrix $M$ contains the set of orthogonal vectors defining the linear measurements, and  $\bar{M}$ contains a set of orthogonal columns that span the nullspace of $M$ (i.e., it is the orthogonal complement of M).
>
> (2) $y$ is equal to $x$ plus independent Gaussian noise.  Thus $y^c$ is equal to $x^c$ plus independent Gaussian noise. So given $x^c$, $y^c$ does not provide any additional information about $y^u$. That is,  $y^u$ is independent of $y^c$ when conditioned on $x^c$.
>
> (3) In our notation, $y$ and $x$ are vectors representing full images (y = x + noise), and are thus always the same dimensionality. The measurements $x^c$, arise from a low-rank matrix $M$, and have lower dimensionality. For example, in inpainting, the dimensionality would be equal to the number of retained (surrounding) pixels.
>
> (4) We referred informally to Eq 2 as expressing a least squares estimate, but agree that it would more properly be called the MMSE solution.
>
> (5) Regarding figure 6: thank you for pointing out the inconsistency in the notation, which we will correct.
>
> (6) One of the main advantages of our algorithm is its efficiency.  We do not rely on Monte Carlo procedures to sample from a fixed density.  Rather, we ascend the gradient of a continuously parameterized family of densities. The convergence relies on some assumptions regarding the denoiser, which (approximately) hold for our denoiser, and which we demonstrate empirically for different parameter settings  (Fig 1).  We also compare run times to other methods for the spatial superresolution example (Table 1), and provide run times for our method across 	all five applications (Table 4).
>
> (7) We were not able to find another reference for DeepRED besides the one we cite (ref [13] - Mataev, Milanfar, Elad, 2019), and its corresponding arXiv version. If we have missed something, we would greatly appreciate the reviewer pointing us to the reference, and would be happy to include it in the revised paper.

---

> > ### Comment · Reviewer_N12S · 2021-08-22
> > **Thank you for your detailed reply**
> >
> > I would like to thank the authors for a detailed reply, which addressed most of my comments.
> > With a better organization, the paper can be in quite good shape.
> > For (7), I mean the reference in the notation of the 3rd column figures for DeepRED should be [13] instead of [7].
> > Also, if possible, could you move Fig. 6 to the main draft instead of a complimentary one? This will make the paper more readable.
> > Overall, I would like to improve my score to 6.

---

### Decision · Program_Chairs · 2021-09-27

**Decision:**

Accept (Poster)

**Comment:**

This paper provides an original way to solve linear inverse problems when given a denoiser.  The method uses forward calls to the denoiser in a way that leverages the signal prior implicit to it.  The paper establishes strong quantitative and qualitative recovery performance.  In the camera ready version of the paper, the authors should make the modifications discussed with the reviewers.  In particular, the authors should clarify the relationship between the proposed method and Langevin Dynamics.  The authors say the method draws "high probability samples"; the authors should clarify what these terms mean and whether or not it is the same as directly sampling from the relevant distribution.